# Virtual Stencil for Patterning and Modeling in a Quantitative Volume Using EWOD and DEP Devices for Microfluidics

**DOI:** 10.3390/mi12091104

**Published:** 2021-09-14

**Authors:** Yi-Wei Lin, Ying-Jhen Ciou, Da-Jeng Yao

**Affiliations:** 1Institute of NanoEngineering and MicroSystems, National Tsing Hua University, Hsinchu 30013, Taiwan; YiWeiLin@itri.org.tw; 2Mechanical and Mechatronics System Research Laboratories, Industrial Technology Research Institute, Hsinchu 30013, Taiwan; 3Department of Power Mechanical Engineering, National Tsing Hua University, Hsinchu 30013, Taiwan; janetciou1022@gmail.com

**Keywords:** EWOD, patterning, droplet manipulate, 3D structure formation

## Abstract

Applying microfluidic patterning, droplets were precisely generated on an electrowetting-on-dielectric (EWOD) chip considering these parameters: number of generating electrodes, number of cutting electrodes, voltage, frequency and gap between upper and lower plates of the electrode array on the EWOD chip. In a subsequent patterning experiment, an environment with three generating electrodes, one cutting electrode and a gap height 10 μm, we obtained a quantitative volume for patterning. Propylene carbonate liquid and a mixed colloid of polyphthalate carbonate (PPC) and photosensitive polymer material were manipulated into varied patterns. With support from a Z-axis lifting platform and a UV lamp, a cured 3D structure was stacked. Using an EWOD system, a multi-layer three-dimensional structure was produced for the patterning. A two-plate EWOD system patterned propylene carbonate in a quantitative volume at 140 Vpp/20 kHz with automatic patterning.

## 1. Introduction

Printing technology has traditionally been widely used to print books, newspapers and printing electronics. Existing printing techniques include inkjet printing [1,2], screen printing, gravure printing, letterpress printing and roll-to-roll printing [3,4]. In 2010, Jinsoo Noh et al. evaluated the limit of accuracy of printing registration of a gravure printing system [5]. In most above methods, the service life of the mold is curtailed because of the contact between a scraper and a master mold. The master mold has also a problem of high production cost and its production is time-consuming; because of the fixed mold, there is a single master mold corresponding to a single pattern, which is unconducive to an initial development of a product. Therefore, a maskless patterning technology will be of great assistance to early-stage R&D of products and prevent master mold degradation. In contrast, a mold that can be digitally patterned will not degrade when used to print on a substrate.

While printing techniques are important, however, a second application area of interest to us is Structural Electronics. Structure Electronics (SE) is a component with conductive trace on the surface of structure. In the present, injection molding, Fused Deposition Molding (FDM) and Stereolithography Apparatus (SLA) are the main solutions for structure manufacturing. After a structure has been formed, the conductive trace can be patterned on its surface by inkjet printing or aerosol jet printing. At present, conductive traces cannot be embedded into the SE component because the structural fabrication and trace patterning occur as serial processes. It is necessary to develop a technology which could combine these two processes and provide advanced electronic prototypes [6]. Actually, the market of structural electronics has rapidly increased to become a multi-billion-dollar business [7,8].

To address applications such as those we have highlighted, we examine the use of a microfluidic technology to manipulate liquids into arbitrary shapes using digitally addressed electrodes. In recent years, microfluidic technology has developed rapidly so as to have become an indispensable part of many engineering and biomedical applications [9,10,11,12,13,14]. Microfluidics is regarded as an important advance in the fields of molecular analysis, biological defence, molecular biology and microelectronics [15,16,17]. Integrating microfluidic components on a single chip becomes a lab-on-a-chip (LOC) [18,19], micro-total-analysis system [20,21], μ-TAS [22] and point-of-care diagnostic equipment [23,24]. With the invention of microfluidics, Shigang et al. integrated EWOD and L-DEP on a single chip in 2006 and used the characteristic that a liquid follows an L-DEP electrode to form the letter pattern “NCTU” [25,26]. In 2013, Fujita et al. proposed a method that can rapidly manufacture microfluidic wafers using dielectric wetted-array wafers. Liquid-phase paraffin served as the medium; ionized water was then used to create a specific shape [27]. The microfluidic patterning was restricted to the designated shape of the electrode; in sum, the patterns of microfluidics were based on the mask [28].

While a dielectric particle was forced by a non-uniform electric field, we call this phenomenon dielectrophoresis (DEP). Even when the particle is not charged, the force still remains. All particles exhibit dielectrophoretic activity in the presence of electric fields. This mechanism is applied in polarized biological objects wildly.

The principles of electrowetting (EW) and electrowetting-on-dielectric (EWOD) are used in microfluidic devices, mentioned in 2002. Kim et al. also conducted interrelated research developed over the years. Several recent commercial applications are based on electrowetting technology, such as adjustable lenses, display technology and much biotechnology [29,30,31,32,33,34]. In 2015, Banerjee et al. proposed a programmable microfluidic system that integrates a continuous fluid and digital microfluidics [35]. Digital microfluidic technology through electrowetting on dielectric (EWOD) systems is widely used, such as in a laboratory on a chip [36,37,38,39]. In 2014, Liu proposed a simple and accurate image-based technique to measure a droplet volume to generate nanometer droplets from continuous electrowetting microchannels [40]. In this work, we used electrowetting-on-dielectric (EWOD) wafers to pattern different kinds of fluid and DEP systems for solidifying the three-dimensional structure stack. We discuss the volume accuracy of droplets generated under various changes of a driving environment such as the number of generating electrodes, the number of cutting electrodes, voltage, frequency and the distance between the two plates. We used an array-type electrode to perform accurate volume patterning. In the part in which the three-dimensional structure is stacked, a mixed colloid of polyphthalate carbonate (PPC) and a photosensitive polymer material was used. The colloid was driven and stacked on light curing to complete the production of the three-dimensional structure.

The purpose of this research was to surmount the limitation that a single mold can correspond to only a single pattern, meanwhile developing a process which could combine structure fabrication and conductive trace fabrication. Finally, this research will realize the concept by the arrayed electrode and curable colloid which were used to complete a single array chip but could be lined with various mold structures. The parallel-plate EWOD and DEP chip can be machined but controlled in volume, with advantages to refine the use of colloids.

Although the electrode design concept and driving force are approximately the same, we drove a smaller liquid volume (3.2 nL for propylene carbonate) and applied UV-curable liquid (UV-SIL) for forming digital patterns in this research firstly. We also make it possible for printing electronics and structural electronics application.

## 2. Theoretical of Electro-Wetting Effect, Experimental Design and Setup

### 2.1. Theoretical

Therefore, a lower limit on the size of a droplet that can be driven is not known and the force applied is simply proportional to the voltage applied. These are important considerations for the application of this approach to high fidelity digital printing at high rates.

Consider a system composed of liquids and solids. When a voltage is applied to the system, the charges and electric dipoles in the system will be rearranged and distributed between the liquid-solid interfaces [41,42,43]. The phenomenon is shown in Figure 1. The surface energy of the interface changes, and the repulsive effect between the charges reduces the surface tension of the liquid. It will result in a smaller contact angle between the liquid and the solid, which in turn changes the wetting characteristics of the liquid on the solid surface. This phenomenon is called the electro-wetting effect [44,45,46,47,48,49,50].

Consider the interface tension of a liquid drop on a solid surface: (1) solid-liquid interface tension **γ_SL_**; (2) solid-gas interface tension **γ_SG_**; (3) liquid-gas interface tension **γ_LG_**. The relationship between them is shown in Figure 2. From Young’s equation, it can be determined that the relationship between the droplet contact angle **θ****_0_** and the tension is:**γ_SG_ = γ_SL_ + γ_LG_ cosθ**(1)

According to the research results published by Lippmann in 1875, the interfacial tension **γ_SL_** between liquid and solid can be controlled by an external voltage, and the derived formula is:**γ_SL_(V)** = **γ_SL_** │ **_V=0_** − **C/2** ∗ **V^2^**
(2)
where C is the capacitance value of the dielectric layer.

Combining the above Formulas (1) and (2), the following can be derived:**cosΘ_V_** − **cosΘ_0_** = **Ɛ_r_Ɛ_0_/2tγ_LG_** ∗ **V^2^**(3)
where Ɛ_0_ is the vacuum dielectric constant, Ɛr is the dielectric constant of the insulating layer and t is the thickness of the insulating layer.

### 2.2. Electrode Design of EWOD Chip and DEP Chip

In the experiment, we used AutoCAD 2019 to draw the electrode designs. There are mask designs of two types for microfluidic patterning and modeling. The first mask design is an electrode array composed of storage tanks, transmission electrodes, liquid-generating electrodes and liquid-pattern 4 × 4 array electrodes. The storage tank electrode is designed with a radius of 2 mm; transmission electrodes are 1.358 mm × 0.762 mm to generate a mother droplet with electrodes of 0.972 mm × 0.762 mm. The patterning area size is 2.55 mm × 2.75 mm; the array is composed of 40 electrodes. The electrode width is 0.15 mm, shown in Figure 3a.

The second mask design is to form a three-dimensional structure; an open interdigitated electrode array chip is designed so that a liquid can be driven without or with an additional upper cover. The architecture contains 7 × 6 sets of interdigitated array-patterned electrodes of width 1.4 mm × 1.4 mm; the overlapping part is 300 μm shown in Figure 3b, which also shows two pairs of electrodes; the interdigitated electrodes are designed with width 20 μm and pitch 20 μm.

### 2.3. Fabrication Process of EWOD and DEP Chip

The EWOD and DEP chip fabrication is shown in Figure 4. The structure of the chip and top plate is indium tin oxide (ITO) glass. The chip is made with photolithography. We chose SU8-2002 as the dielectric layer and Cytop for the hydrophobic layer.

### 2.4. Experimental Setup

The digital microfluidic control system used in the experiment includes a signal generator, power amplifier, relay board and PXI-6512. The electric signal was generated with a signal generator (33220A, Keysight., Santa Rosa, CA, USA) and magnified with a power amplifier (A304, A. A. Lab System Ltd., Ramat Gan, Israel). This electrode signal was connected to the relay board. To control each relay port, we used PXI-6512 (National Instruments Corp., Austin, TX, USA), which has 64 channels and can control I/O switches in a safe output status. Through LabView program and the clamp (CCNL050–47-FRC, Yokowo Co., Ltd., Tokyo, Japan), we controlled specific switches and electrodes. The signal was output to the chip by the clamp.

### 2.5. Preparation of Driving Liquid

During the patterning and modeling experiment, we selected propylene carbonate (Sigma-Aldrich, St. Louis, MI, USA) and UV-SIL photosensitive polymer material (UV 329, Heart-bond industrial materials LTD., Changhua County 50077, Taiwan) as the driving liquid. UV-SIL was selected as the modeling experiment material and absorption wavelength was 365 nm, because UV resin is an adhesive material of a kind that can become solidified on exposure to UV light, and it is commonly used as a glue for paints, coatings and inks.

## 3. Results

### 3.1. Droplet Generation

This experiment used a dielectric wetted 4 × 4 array electrode wafer to control the droplet; the liquid used was propylene carbonate (PC). The distance between the parallel plates was defined with double-sided tape as 20 μm. The applied voltage was 120 Vpp; the frequency was 20 kHz to make the mother droplet become generated from a storage-tank electrode. The four basic operations of droplets on applying a voltage mentioned in the literature review are manipulation, generation, transmission and separation [33]. Accurate control of the electrode switch was achieved through a computer program (LabView).

Initially, liquid (1 μL) was placed on the electrode of the storage tank. After the top plate was covered, a mother droplet was generated through the steps of transmission and separation. The area of the mother droplet after generation was 25,000 μm^2^; the volume of the mother droplet was 0.016 μL, as shown in Figure 5a. Subsequent droplets were generated from the mother droplet (0.016 μL) to 1.95 nL. The liquid-generation electrode was designed to be 0.2 mm × 0.4 mm to separate the droplet. The mother droplet electrode was first actuated, followed by the cutting electrode and small transfer electrode; the size of the droplet was similar to that of the small transfer electrode. After all three electrodes were actuated, the mother droplet moved only the same distance as the opened electrode. Instead of moving the small transfer electrode completely, it exhibited a state of force balance. The cutting electrode was then closed to generate a droplet; the droplet was generated according to steps shown in Figure 5b.

### 3.2. Droplet Generation in a Two-Plate Chip

In the experiment to generate a droplet of quantitative volume, with the variation of the number of electrodes, we observed the changes in the size of droplets. We defined the gap between the two plates to be 20 μm; the fixed frequency was 20 kHz, and varied voltages were applied. A droplet was generated from the mother droplet; the mother droplet was stretched to three, four and five electrodes; a cutting electrode was then used to generate droplets, as shown in Figure 6a. The solution results show that, when a larger voltage was applied, the separated droplet was also larger; the further that the mother droplet was stretched, the larger was the droplet produced. In the experiment, when the applied voltage exceeded 200 Vpp, the liquid stretched along the electrode wire, as shown in Figure 6b. The experimental results of droplet size with varied voltage are shown in Figure 6c. The correlation coefficient between the size of the droplets generated with three electrodes and the voltage was R_2_ = 0.991.

In other experiments with the number of cutting electrodes, the mother droplet was separated into droplets, using 1, 2 and 3 cutting electrodes, respectively, and generated ten times to obtain an average value. Fewer cutting electrodes caused a less dragging tail so that the size of the generated droplet was more consistent with the electrode size. The dragging of the liquid tail is shown in Figure 6d. Fewer generating electrodes and shorter cutting paths were observed to generate a smaller droplet and were near the designed electrode size, on comparing the change of frequency from 10 kHz to 30 kHz. Figure 6e shows the area of the droplets generated using various generating electrodes and cutting electrodes at 120, 130, 140 and 150 Vpp and varied frequency. The size of droplets generated was unaffected by the frequency.

### 3.3. Variation of Droplet Size with Gap Height and Voltage

The feasibility of forming a droplet was established through the above experiments. In this experiment, we tested the generation of droplets with varied gap height under the same frequency, 20 kHz, and a varied voltage level. The purpose was to find a suitable gap size to facilitate subsequent patterning to define a required liquid volume. In this experiment, 11 voltages/Vpp = 100, 110, 120, 130, 140, 150, 160, 170, 180, 190 and 200, were applied at fixed frequency 20 kHz. The size of the droplets generated in this experiment was averaged to determine the size of the droplets generated by the voltage. We used spacers of thickness 20, 15 and 10 μm to define the gap height of the two-plate EWOD system. For droplet measurement, we used an image analysis tool to obtain the size of the droplet generated from a pixel calculus. The calculus equation is
measure pixel × scale^2^ = droplet area

The result of varied gap height on the generated droplet size and area is shown in Figure 7a; the error axis is the error value of five experiments. A height of 10 μm was more appropriate to generate droplets because the size of the droplets was stable and consistent with the designed electrode size; the error of the generated droplet from ten droplets is shown in Figure 7b; and the droplet size generated by the electrode design must be about 80,089 μm^2^.

As the results show above, droplet size could be control by gap height and applied voltage. However, the smallest droplet size was still restricted by electrode size. This means that with reducing the electrode size, the smaller and finer features will succeed.

### 3.4. Microfluidic Patterning of 4 × 4 Array EWOD Electrodes

Based on the above experiments, we used three generating electrodes and one cutting electrode to generate droplets. A spacer (10 μm) defined the gap between the parallel plates. The experiment used propylene carbonate to pattern various letters to confirm the amount of liquid required for each letter. At 140 Vpp/20 kHz, the size of the droplets generated in an environment with gap height 10 μm was about 0.83 nL. The experiment patterned 26 alphabetic characters. The statistical results of the size of droplets required for each letter are shown in Table 1. The pattern required two more droplets than the number of electrodes because of the closed structure, such as A, B, D, P, Q, shown in Figure 8a. This quantitative method of droplet generation can accurately determine the amount of liquid required, which was conducive to the volume of liquid needed for patterning such as for future molding. An appropriate volume of liquid can make the shape of the molded shape nearer the original. This design was less likely to cause a problem of shape deformation.

Next, we used a pre-editing method to automate the patterning. The strokes of each pattern varied; the total volume also varied. We compared the interval spent by each pattern in the automated molding; the results are shown in Figure 8b. To turn on the pattern of the synthetic electrode takes longer; the time to form a letter with a higher pattern continuity or fewer strokes is shorter. The quantity of opening electrodes and droplet volume needed pattern an English alphabet is shown in Table 1.

As the results show above, it could take an electrode from each EWOD chip as a pixel of display. Higher resolution of display means with a higher amount of pixels, it could show finer, sharper characters in the image, even if a higher resolution of electrode design on EWOD chips in this research was not shown. A high possibility and potential to address the requirement for different shapes still remains if a high-resolution electrode of EWOD chips is designed and fabricated.

### 3.5. UV-SIL Photosensitive Polymer Colloid Patterning Experiment by EWOD Electrodes

Based on the foregoing experiments, three generating electrodes were used for patterning experiments; one cutting electrode was used to generate droplets, and a 10 μm spacer served between parallel plates. The experiment used UV-SIL to pattern various letters to confirm the amount of liquid required for each letter. A photosensitive material has the characteristic of changing from a liquid to a solid in a short time after absorbing ultraviolet light. A fixed voltage, 120 Vpp, and fixed frequency, 20 kHz, were applied; an area of 2 × 2 was used for letter patterning on the 4 × 4 array electrode. The size of the droplets generated in an environment with a gap height of 10 μm was about 0.8087 nL. Directly driving the UV-SIL liquid in the parallel plates facilitated direct curing when reprinting in the future. The curing of the colloid required two stages, pre-curing and full curing, when the electrode was open. The upper cover must be added between curing to complete the reprint. Compared with an open-type coplanar electrode, the two-plate electrode did not need to be covered with additional pressure, so it is less likely to cause shape deformation. The results of the volume of liquid used appear in Table 2.

### 3.6. Forming a 3D Structure Using Stacking with UC-SIL Photosensitive Polymer Colloid by DEP Electrodes

In the UV-SIL modeling experiment, we used a 7 × 6 open-environment electrode to control the liquid, placing 1.5 μL of the colloid on the electrode. With a voltage of 360 Vpp and a frequency of 5 kHz, the colloid completely covered the electrode in two seconds, as shown in Figure 9a. When the three-dimensional structure was stacked, the pattern was easily formed according to the shape of the first layer structure. The two-plate DEP system was combined with open-environment electrode patterning reprinting to make the first layer shape clearer. The first layer was patterned and formed with a combination of parallel-plate type and interdigitated electrode. We used the original 7 × 6 interdigitated electrode, with 10-μm spacers, and chose the upper cover with a hydrophobic layer.

After the UV-SIL formed in the parallel plate, it was irradiated with a UV mercury lamp (350 nm–450 nm) for 9 min. The top cover of the hydrophobic layer was removed; the formed pattern was printed on the top cover without the hydrophobic layer using the same material as an adhesive, finally curing with UV light for 9 min that allowed the liquid to solidify and to be transferred to the upper cover without a hydrophobic layer. If a narrow band of wavelength with a higher dosage UV LED is used, shorter curing time will be possible. The same chip was subsequently used to pattern the electrode-driven colloid in an open environment. After forming, the upper cover with the first layer of colloid was used for imprinting. In this experiment, we found that the glue on the electrode chip was well integrated with the glue on the upper cover; through the height control with the Z-axis lifting table, the stacking and mold turning steps were completed smoothly. The process is shown in Figure 9b.

During the stacking, when the first layer structure was completely defined, whether the subsequent layers were driven with an electrode to drive, the liquid had little effect on the molding. In the experiment, the first layer mold structure was 10 μm, in which the colloid was formed in the parallel plate; the structural forming of letters T, U, H is shown in Figure 10a. During the stacking, the colloid was placed directly on the chip with the hydrophobic layer; with no applied voltage, the embossing was performed on controlling the lifting height to complete the stacking of the three-dimensional structure. The result of the stacking structure is shown in Figure 10b; the transfer error between each layer is shown in Figure 10c. With satisfactory control of the stacking height, the transfer error between each layer was less than 10%.

## 4. Conclusions

We discuss various methods to generate a more accurate volume of the droplets, to obtain improved environmental parameters for accurately generating droplets. In the subsequent patterning experiment, an environment with three generating electrodes, one cutting electrode and a gap height of 10 μm was used. The size of the droplets generated was relatively stable and consistent with the designed electrode size. In the experiment of propylene carbonate and UV-SIL patterning, the applied voltage was 140 Vpp/20 kHz, and generation, movement, separation and positioning were performed; the letters were patterned. Through pre-programming we can also perform patterning automatically.

In the modeling experiment, a new driving liquid called UV-SIL colloid was used, better to control under 360 Vpp/5 kHz. We formed the first layer with a two-plate EWOD system to define the pattern; after UV light curing and stacking for 9 min, the colloid was completely transferred to the upper cover with no hydrophobic layer. The most important factor in the experiment was the height control. During the experiment, the subsequent layers of colloids became fused along the defined edge of the first layer.

Compared to common printing techniques in Figure 11a, a master mold is necessary for all printing techniques. A blading process is used for ink feeding within contact between the blade and the master mold (red dotted circle in Figure 11a). This process will cause a risk of damage in the surface of the master mold, which results in a limited life span of the master mold. Another issue is that only one pattern corresponds to its master mold. In this research, we successfully show digital patterning by a DEP system in Figure 11b. Without the blading process and variety patterns in one digital mold (DEP system), printing is made more cost-effective and efficient for the development stage.

We successfully demonstrate a new concept of a 3D printing technique for structural electronics application. Through such EWOD and DEP systems, new UV-SIL material were introduced, and the 3D stacking process was approved by UV exposure. A clearer pattern outline and a 3D multilayer solidified stack with less deformation were thereby obtained. Compared to traditional printing techniques, the EWOD and DEP system in this research introduced that it is possible to pattern without master mold damage. On the other hand, 3D printing through this DEP system also allows new comparisons in the literature. It makes an embedded circuit for structural electronics possible, which is impossible for current 3D printing techniques.

In the future, if more functional driving liquids are introduced and approved in this DEP system, more interesting electronic products will be possible. The concept is shown in Figure 12.

## Figures and Tables

**Figure 1 micromachines-12-01104-f001:**
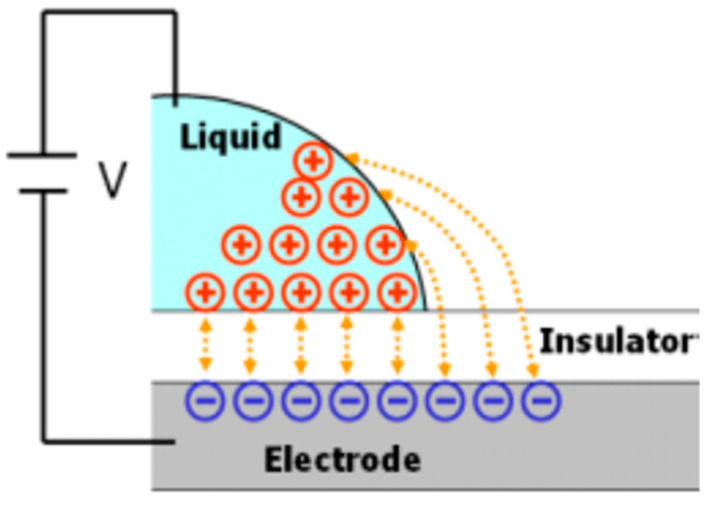
The charge is rearranged according to the applied voltage.

**Figure 2 micromachines-12-01104-f002:**
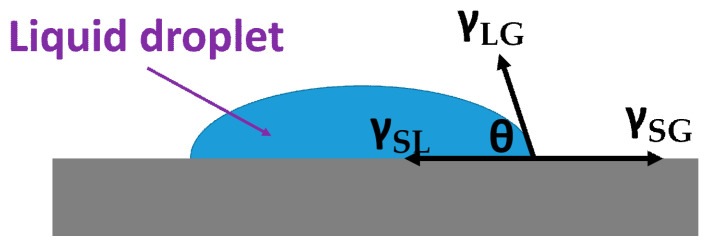
Interfacial tension distributed.

**Figure 3 micromachines-12-01104-f003:**
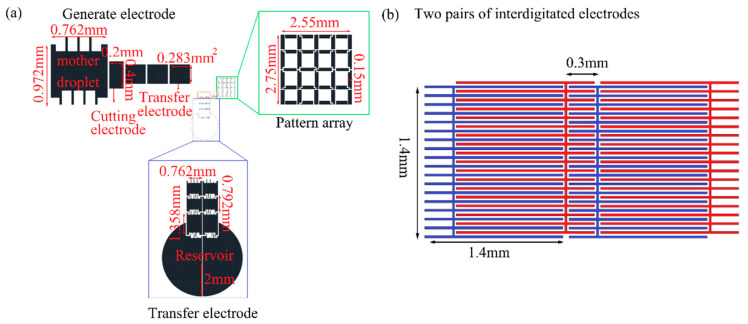
Chip designs: (**a**) electrode mask for an EWOD pattern array; (**b**) two pairs of electrodes in 7 × 6 open interdigitated electrode mask for DEP chip.

**Figure 4 micromachines-12-01104-f004:**
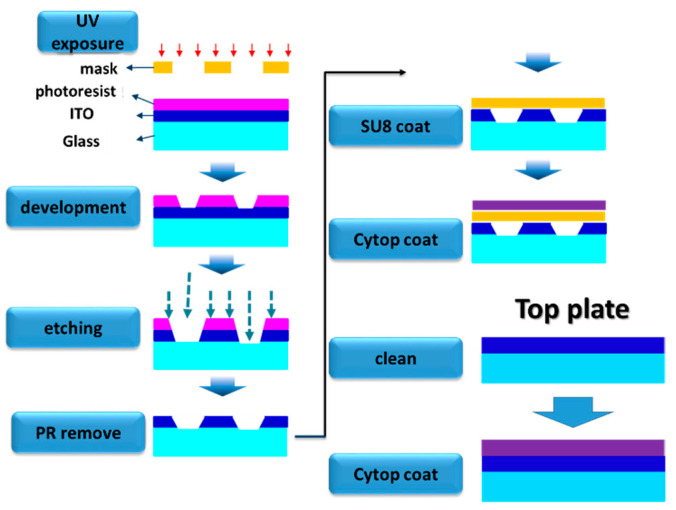
EWOD and DEP chip fabrication flow.

**Figure 5 micromachines-12-01104-f005:**
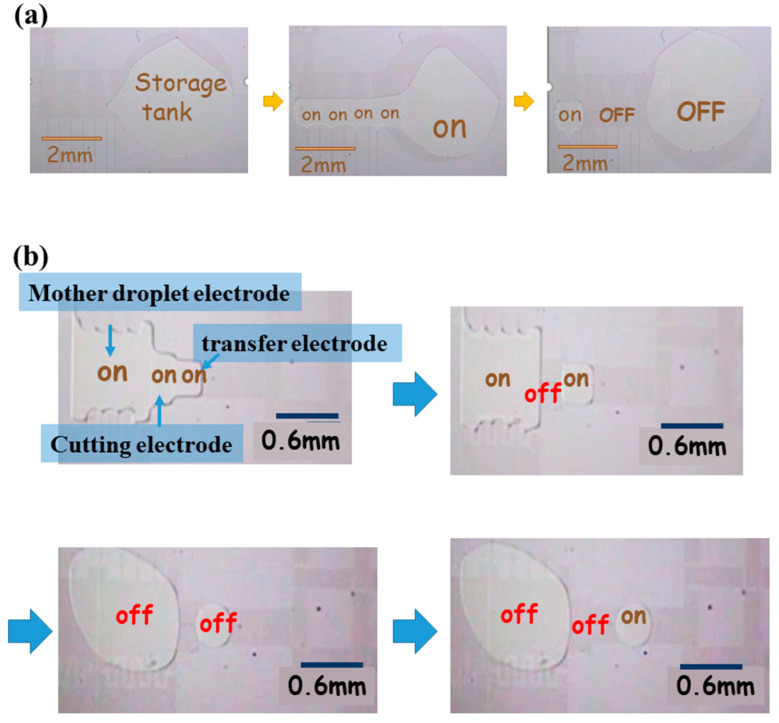
Droplet generation; (**a**) mother droplet generation; (**b**) droplet generated and moving forward from a mother droplet.

**Figure 6 micromachines-12-01104-f006:**
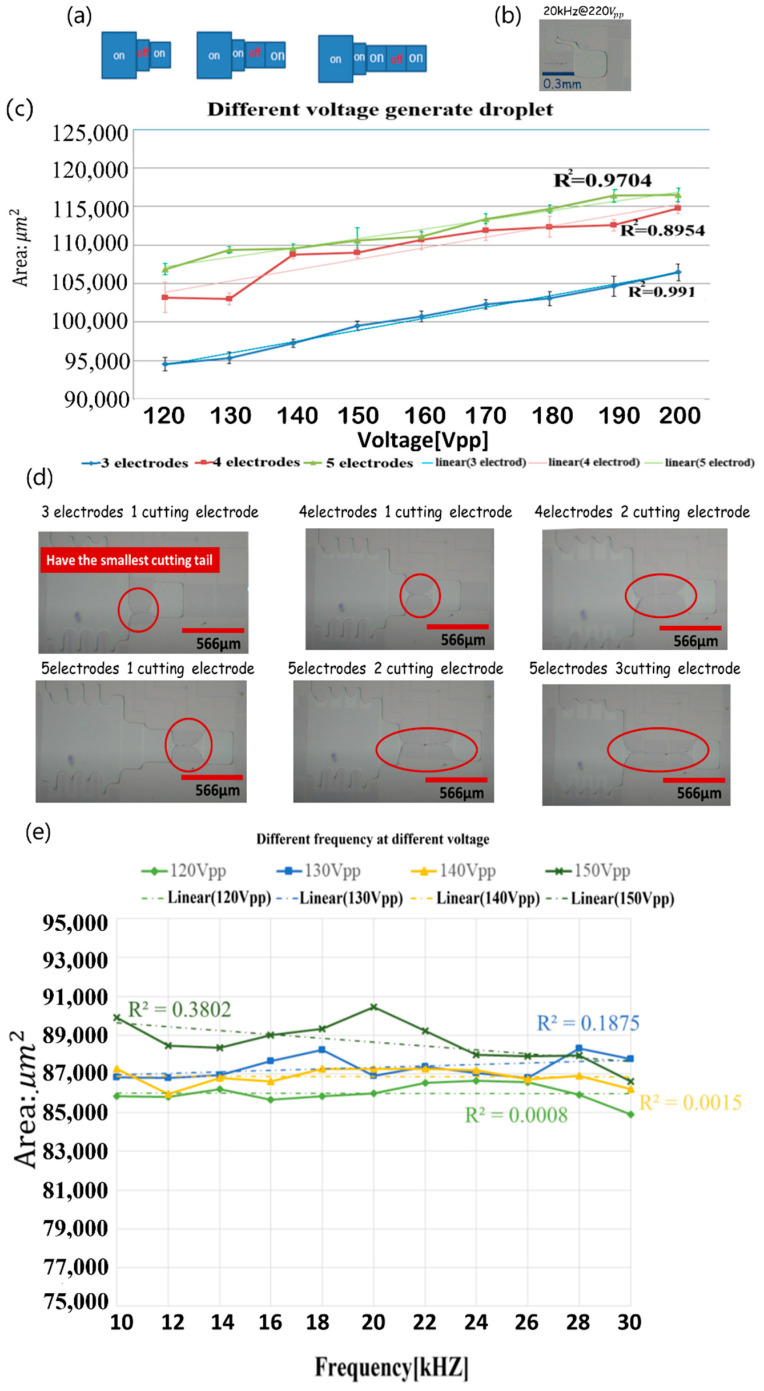
(**a**) Droplet generation with 3 electrodes, 4 electrodes and 5 electrodes; (**b**) liquid stretched on a wire at 220 Vpp; (**c**) variation of droplet size with voltage and number of electrodes; (**d**) dragging of the liquid tail phenomenon with different electrodes and cutting electrodes; (**e**) variation of droplet size with frequency at 120 Vpp, 130 Vpp, 140 Vpp and 150 Vpp.

**Figure 7 micromachines-12-01104-f007:**
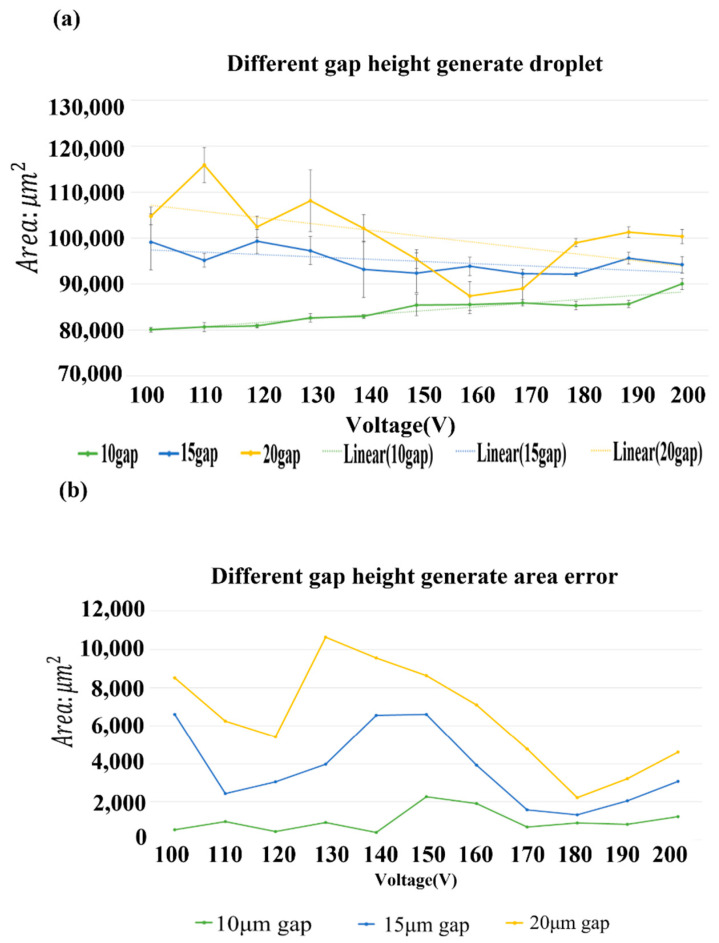
(**a**) Variation of droplet size with 10 µm, 15 µm and 20 µm gap height and voltage variety; (**b**) error of area measurement with 10 µm, 15 µm and 20 µm gap height and voltage variety.

**Figure 8 micromachines-12-01104-f008:**
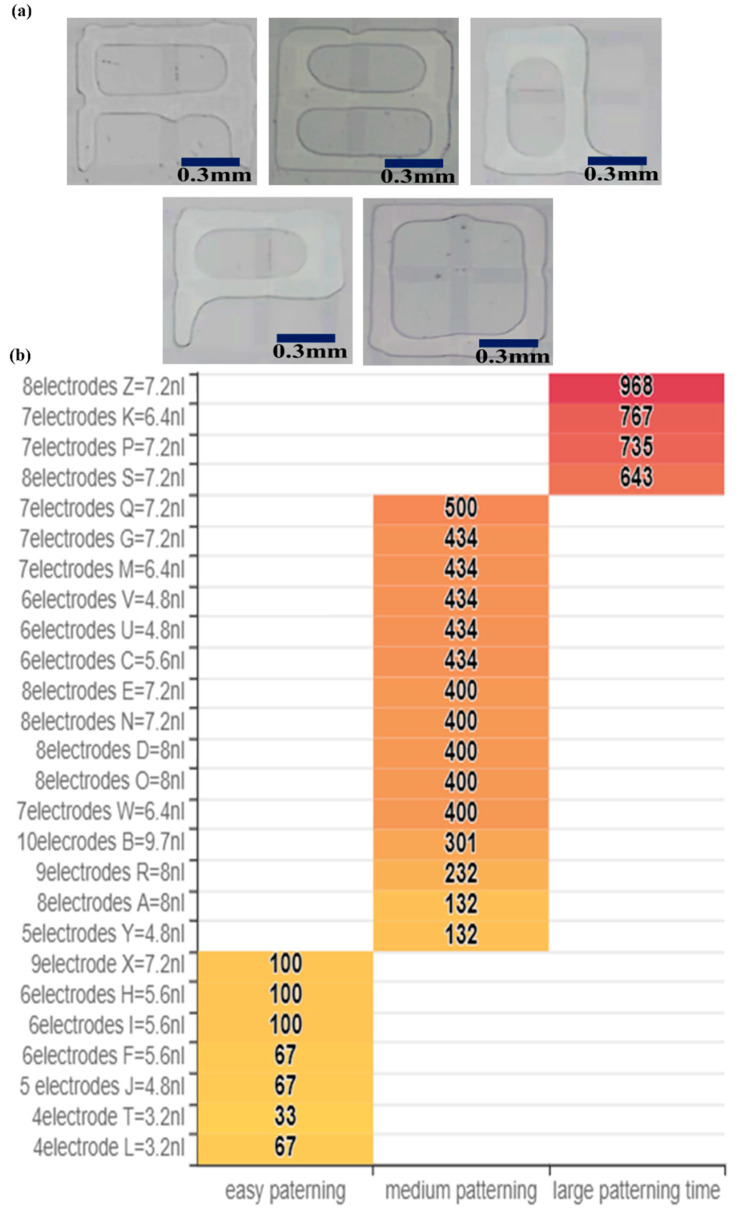
(**a**) Microfluidic patterning, A, B, Q, P, O at 140 Vpp/20 kHz and with 10 µm gap height. (**b**) Amount of time and volume taken to form the letters A to Z.

**Figure 9 micromachines-12-01104-f009:**
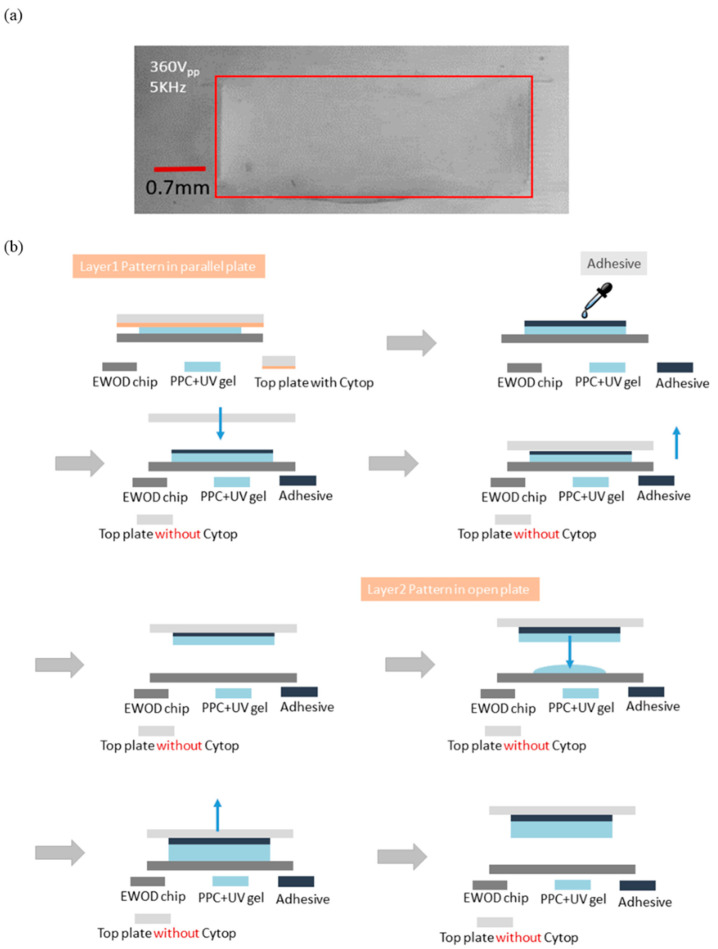
(**a**) 360 Vpp/5 kHz colloid completely covers electrodes in two seconds; (**b**) model stacking process flow by DEP system with UV-SIL liquid.

**Figure 10 micromachines-12-01104-f010:**
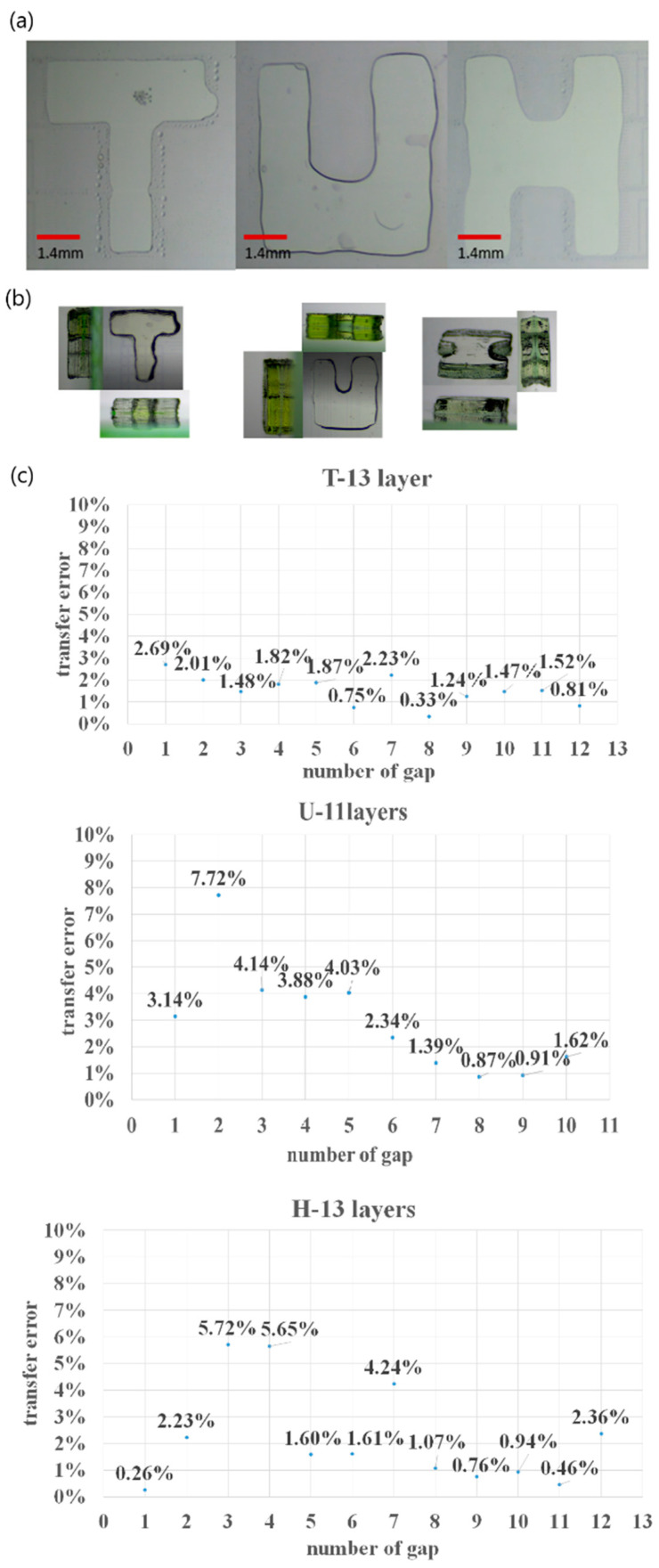
(**a**) First layer structure forming by DEP system, T, U, H; (**b**) 3D stacking structure result of T, U, H with 13 layers, 11 layers and 13 layers, respectively; (**c**) transfer error between each layer during stacking.

**Figure 11 micromachines-12-01104-f011:**
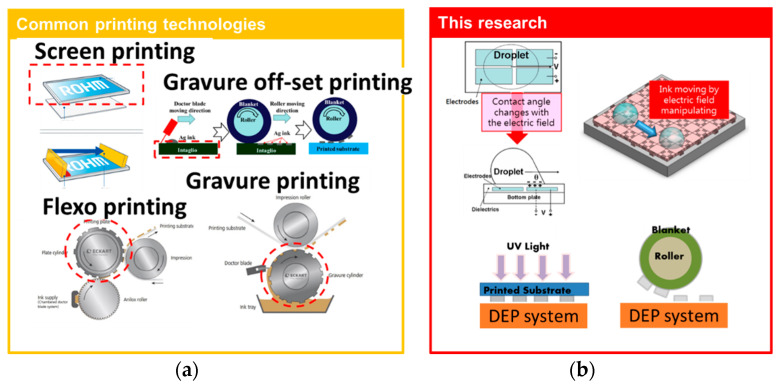
(**a**) All printing techniques with blading process and pattern-fixed master mold; (**b**) digital mold by DEP system without the blading process.

**Figure 12 micromachines-12-01104-f012:**
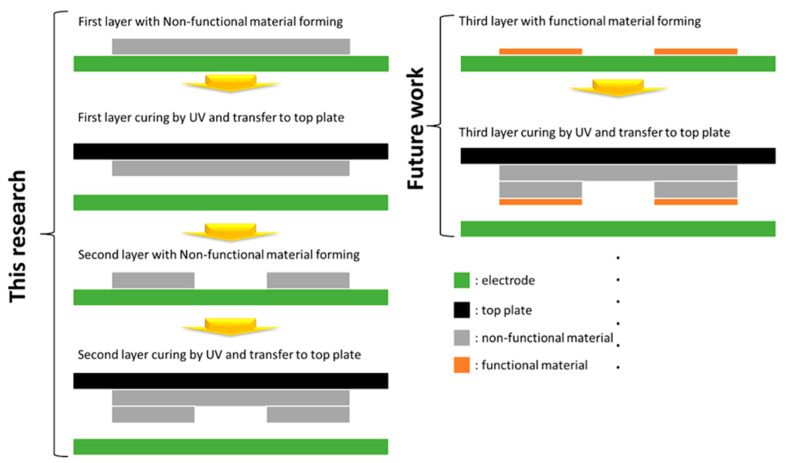
Current 3D structure fabrication flow in this research and application in functional material with DEP systems for structural electronics in the future.

**Table 1 micromachines-12-01104-t001:** Quantity of opening electrodes and droplet volume to pattern an English alphabet.

English alphabet	A	B	C	D	E	F	G
Electrode number	8	10	6	8	8	6	8
Droplet volume (nL)	8.30	9.96	5.81	8.30	7.47	5.81	7.47
English alphabet	H	I	J	K	L	M	N
Electrode number	6	6	5	7	4	7	8
Droplet volume (nL)	5.81	5.81	4.98	5.81	3.32	6.64	7.47
English alphabet	O	P	Q	R	S	T	U
Electrode number	8	7	7	9	8	4	6
Droplet volume (nL)	8.30	7.47	7.47	8.30	7.47	3.32	4.98
English alphabet	V	W	X	Y	Z		
Electrode number	6	7	8	5	8		
Droplet volume (nL)	4.98	6.64	7.47	4.98	7.47		

**Table 2 micromachines-12-01104-t002:** Quantity of opening electrodes and droplet volume to pattern an English alphabet by using UV-SIL liquid.

English alphabet.	A	B	C	D	E	F	G
Electrode number	8	10	6	8	8	6	8
volume (nL)	8.08	9.70	5.66	8.08	7.27	5.66	7.27
English alphabet	H	I	J	K	L	M	N
Electrode number	6	6	5	7	4	7	8
volume (nL)	5.66	5.66	4.85	6.46	3.23	6.46	7.27
English alphabet	O	P	Q	R	S	T	U
Electrode number	8	7	7	9	8	4	6
volume (nL)	8.08	7.27	7.27	8.08	7.27	3.23	4.85
English alphabet	V	W	X	Y	Z		
Electrode number	6	7	8	5	8		
volume (nL)	4.85	6.46	7.27	4.85	7.27

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
