# Peer review of "Virtual Stencil for Patterning and Modeling in a Quantitative Volume Using EWOD and DEP Devices for Microfluidics"

_micromachines, 2021, doi:10.3390/mi12091104_

Round 1

Reviewer 1 Report

The manuscript reports on a two-plate digital microfluidic system capable of precise volume control and deposition of photosensitive polymers to create solid multilayer patterns. The manuscript definitely has merit and the works shows good potential, but the presentation needs to be significantly improved so the most diverse set of readers may also understand the work without difficulty. 

Introduction, conclusion:

  • The extent of the novelty of your work is currently not possible to determine accurately. Previous works are introduced, but a more quantitative comparison to your work would be helpful. In particular, precision of volume control in recent literature. 
  • Digital microfluidics is a highly active field. Therefore, to place your work accurately, upwards of 30 references are needed (ideally 30-50). Consider such questions as: is your electrode layout or layer structure novel? How do key performance metrics of your system compare to other similar works (the state-of-the-art)? If your work is completely unique, it must also be visible from the introduction/conclusion. 
  • Conclusion: you must detail the applicability of your work in research/industry. Are you planning to use this system to create precision molds? Or perhaps even to create microfluidic channel layouts for continuous flow reactors? It is currently unclear from the manuscript. For an expert the potential is clear, but for a non-expert it is not. 
  • Generally, the conclusions should be extended to reflect on all numerical results and compare them to the state-of-the-art introduced in the introduction. 

 Structure of section 2-3:

  • Most of the figures and graphs must be significantly improved for readability and clarity. This includes captions. Particularly, graphs in figure 4-8 need a higher resolution, a longer caption explaining what we can actually see (so a reader can quickly understand before reading the main text for details). Units are sometimes missing or are strangely formatted, figure 7 has Cytop marked as a grammar error, figure 6 caption does not mention "A", figure 5: 10 gap, 15 gap, 20 gap... and so on. Image resolution should also be much better. 
  • Please follow the IMRAD format. I.e., instead of having the methodology as well as the results all in section 3, have the methodology (the experimental plan) for each subsection in section 3 explained in section 2 under subsections. To an expert the experimental plan is clear, but again, to a non-expert, this will be highly confusing unless clearly broken up into methodology and results. 

There are additional minor corrections needed:  

  • Figure 2: PR acronym not introduced, PR resist is doubling
  • ln 122 brand & P/N for double-sided tape needed
  • ln 125 reference to review missing

Reviewer 2 Report

In this report, authors presented virtual stencil for patterning using a microfluidics system. After carefully read and evaluated the manuscript, I recommended the manuscript should be Rejected because of some reasons below:   

  1. The authors claimed that they reported using EWOD device for virtual stencil, but actually, they were using a DEP system. This is very confusing to readers.
  2. The introduction section was generally poor in writing without logical and guiding readers to what they want to cover in their manuscript. For example, Paragraphs 1 and 2 of the Introduction section were off the topic.
  3. Data presentation and figure preparation were not well organized. For example, some letters and language presented in the figure are inappropriate to read (Ex: Fig. 4). The authors tried to show too much data in the figures that make it is not concise.
  4. A similar work had been reported at AIP Advances 10, 125115 (2020); doi: 10.1063/5.0012684.

Reviewer 3 Report

The authors have done a wonderful job of presenting a well-controlled experimental system using a combination of parallel plate EWOD and open face EWOD for layer-by-layer fabrication of colloidal/UV curable photopolymers. The authors have described the fabrication methods in detail. The figures are well laid out and reasonably professional but require some modification for legibility.

The introduction details the prior work and background of the technologies used. The authors promise to address challenges in fabricating molds that are expensive to fabricate and rapidly damaged during printing processes. As such their hope is to rapidly fabricate molds by demonstrating the principal idea of layer-by-layer 3D printing using an EWOD. After the introduction the manuscript describes the fabrication process of the system and alludes to prior work for specific details on modeling the performance of individual EWODs. It is noted that no theoretical or engineering basis is provided for the experiments shown. 

Characterization of the device appears to be a thorough optimization at the length scales used. The key contribution may be the combined use of a parallel plate EWOD and an EWOD used in an open frame system. This enables subsequent layers near the electrode to be formed in a layer-by-layer fashion. Each parameter is iterated including gap height, voltage, frequency, number of electrodes (by type) to generate A-Z. Generally, fewer electrodes enabled more accurate and faster response. The results appear valid and accurate. They are also mostly straightforward to understand and interpret.

However, several issues hold the manuscript back from its potential. These are detailed below:

  • Motivation
    1. The motivation outlined in the introduction is disconnected from the results. It is unclear if the authors imagine this approach may solve master mold fabrication or SE development. If so, please address how in the discussion. In particular, the paragraph detailing structural electronics seems irrelevant as written. A sentence after line 45 may clarify how your technique may address 3D printing challenges in this space. As written, I can’t deduce if the author intends to fabricate electronic devices by using conductive polymers or to capture components? Please clarify. It may not be worth mentioning SE because of these complications.
    2. Naturally, 3D printed molds may be faster and cheaper than machined molds for short runs and prototypes. The approach could be an obvious technique, but it is unclear how this would be implemented. Could this approach work for making a cylindrical master mold? How do you address the requirement of different letters and shapes to have varying numbers of electrodes?
    3. Please differentiate this from other available 3D printing techniques (DLP, laser, LCOS, holographic). These have been shown to do the same already and can achieve very high fidelity. However, the uniqueness of constraining the liquid to a region and then irradiating the entire space is fascinating and could prove useful in unique situations. What might they be!
  • Manuscript structure
    1. Please expand the discussion/conclusion of the paper to enable the reader to appreciate how this relates to the introduction. You could answer if this approach may solve the challenges of fabricating a master mold and how by means of a figure even if the experiment is not performed.
    2. Additional discussion detailing the benefits and limitations of this approach would be considered of significant use to the readers. For example, a 9-minute curing time sounds excessive. Is this because the UV curable polymer was diluted to avoid increasing the viscosity of the liquid? If so, this would be an important consideration. Or was it because you used a weak UV lamp, or the photo-initiator is not well aligned to 365nm UV light? Is there a theoretical limit on the smallest feature that can be fabricated with an EWOD? Were any prior models used to establish the physics of this method? If so, which ones specifically (although I recognize prior work has been cited)? These types of details are missing.
  • Figures
    1. It is unclear if the trend line in Fig. 4c is a model or a fit. If the line is a model, please indicate where the model can be found. If it is a fit, please enlarge the R2 font size.
    2. The font size in the figures (1,4,6,8) falls below the minimum readable size.
    3. Fonts between table 1 and table 2 also vary.
    4. Figure 5b can be inferred from 5a. Minor point, didn’t impact decision.
  • Methods/Models
    1. In the discussion, please indicate what drives this phenomenon shown in Figure 4a-c if known. For example, what is already known about how electrode size impacts transfer rate. If two  electrodes cover the same volume as 8 electrodes, is this the reason the Z takes longer to pattern? i.e. smaller electrodes produce less effect so it takes longer.
    2. As shown in Figure 7 if I count from top left to bottom right (numbered a, b, c, etc.) beginning in f it is unclear how much liquid is transferred. The ‘correct volume’ of liquid would result in constraining the layer to the shape regardless of the EWOD being used due to surface tension. So, it isn’t clear the EWOD is useful after the first layer. A more interesting result to me would be altering the shape between layers. As shown, it is unclear how much the EWOD open plate design is contributing to the shape. If this isn’t the case, please explain. An important control experiment would be to use the double layer EWOD to produce the first layer then place a droplet of appropriate size and lower the first layer onto it and cure it in place. This compared to using the single layer EWOD would give a sense of the role of surface tension. (or a more interesting 3D printed part could be made)
    3. If 9 electrodes are required to produce an ‘X’ then, is a limitation when writing multiple letters that all the letters require 9 electrodes. If so, then is it true all the letters would be limited by the 9-electrode performance? Another approach that could be suggested is a step and flash approach in which a few electrodes are used to make a shape and then the electrodes are moved using a stage to expand add to the shape enabling arbitrary shapes. This of course would be slower but may enable some flexibility.

My recommendation is that this paper undergo major revisions and then be re-submitted for evaluation. The text requires significant expansion in the discussion and clarification of both advantages and limitations. Finally, the take home message is unclear because the 3D printing experiment the paper relies on demonstrates a layer by layer extrusion that may be driven by surface tension. A control experiment demonstrating changing layer shapes between layers may alleviate this concern (or as described in point 2). Otherwise, I'm excited about the work and look forward to reviewing it again in a positive light. 

Round 2

Reviewer 1 Report

Thank you for responding to my queries. However, some of the responses either do not address my concerns or are not reflected in the text. Several of the issues I raised previously still stand:

Introduction: A more quantitative comparison to the state-of-the-art is still missing. I would like to see a table comparing the performance of your system (your novelty) to similar systems demonstrated in recent literature - particularly volume control precision, since this is a vital aspect of electrically controlled wetting and digital microfluidics. Perhaps there are other additional performance metrics where you do better than others. I understand that your primary novelty is dynamic mask generation, but you are not the first group to do this. Therefore, it is vital to have a comparison to highlight your novelty. E.g. how does your work compare to that of Shigang et al.? How is it better? What did you do differently? Even a cursory look at that publication reveals a concerning amount of similarity. This must be addressed. Also, most of the introduction is still not sufficiently relevant to what your manuscript focuses on. I also do not see the relevance or need of the added paragraph between ln 34-42, but I assume it was in response to another reviewer's query.  

Number of references. In your response you stated that you added references up to 50 references. Technically, 33 is between 30-50, so that would be OK. However, the comparison to SOA (state of the art) is still missing (see my previous point above). Plus, most references are too general and/or unrelated to specifically your work. The introduction must not be a historical overview but rather an overview of what's been done in your specific field in the last 5 years and how can we the readers place your work against this background. 

The restructuring to the IMRAD format was not successful unfortunately. Please consult with a supervisor or a senior colleague to help with the restructuring. At the moment the details about the experimental methodology and the setup are still not physically separate from the results. Section 2.8 is for instance, a mixture of both. Whereas the graphs and the corresponding text belong under section 3. The figure captions would still need to contain more details so that the reader can look at them and understand the contents at a glance without having to read into the body of the text. 

Grammatical and style errors are still an issue that needs to be addressed. 

Reviewer 2 Report

(1)  When I first reviewed this manuscript, the main concern that led me to recommend the journal to “Reject” in the first round was that I was not convinced they used the EWOD system as stated in the manuscript. I pointed out that their system was DEP (Dielectrophoresis) which operated using dielectrowetting to generate liquid-dielectrophoresis force to change the contact angle of a droplet. Of course, EWOD and L-DEP systems must be coated hydrophobic-dielectric layer. However, EWOD and L-DEP systems are different such as electrode design, liquid droplets (conductive, non-conductive).

What convinced me that the system is L-DEP are (i) the authors used dielectric liquid (Propylene carbonate liquid) and (ii) electrode design (interdigitated electrodes). Interdigitated electrodes could lead to generating the non-uniform electric fields (L-DEP force).  

Would you consolidate again with these Ref.: Lab Chip, 2017,17, 1060-1068; https://doi.org/10.1039/C7LC00006E and AIP Advances 10, 125115 (2020); https://doi.org/10.1063/5.0012684.

(2)  Introduction was generally improved. However, Paragraphs 4 and 5 must be revised from EWOD focus to L-DEP focus to make logical flow.

(3)  “(from Industrial Technology Research Institute, ITRI)”, this information of material should not be written in the abstract. It should be moved.   

Reviewer 3 Report

I appreciate the authors response and alterations to the manuscript. However, I would regard the alterations as minor not major. Moreover, they have introduced new issues and failed to address several concerns within the text.

The introduction and conclusion do not clearly describe how this device solves the problem of either structural electronics or master mold fabrication beyond a superficial sense of forming polymer shapes. The reader cannot deduce if the author wants to fabricate molds using this method or use this method as a digital mold. What I mean specifically, is if the electrodes are used to fabricate a mold, that mold would be no more adjustable than a standard mold. If on the other hand the author means that a large array of electrodes would replace an engraved mold for printing colloids that is not communicated.

Regarding structural electronics, it isn’t clear how this solves trace fabrication inside of a device. Is the intent the colloid could be conductive?

In the future I would select one of the two problems raised as the motivation statement and use the extra space to provide a clear diagram imagining how this would eventually be used. If this is not desirable then perhaps a different, simpler motivation is needed that doesn’t distract from the science at hand.

The added text contains many new structural and grammatical errors.

Scientifically, I appreciate the demonstration using photocurable materials with an EWOD. The author has kindly added the theory behind electrowetting which is appreciated but disregards what I asked regarding theoretical models of size and speed. A number of publications have been produced detailing forces applied to liquid drops in EWODS. This would be much more relevant considering all the measurements they present discuss size, volume and rates. This is more confusing due to Figure 9, missing units. Is this milliseconds? If so, that’s an opportunity to really drive home the advantage of this method. 

Permissions to include the images not owned by the authors in Fig. 3 and Fig. 2b. These have been previously published and therefore require permissions from the publishers.

While I didn’t indicate it in my review specifically, it should be obvious that Figure 11 c suffers from the same text issues as the other figures I highlighted. 

In summary, I still regard this as important, interesting and valuable work. However, as written it lacks clarity, consistency, and logical structure. I think a much simpler, less ambitious motivation with a better context will aid the manuscript. I'm also still not convinced the final figure shows that the EWOD is driving the shape formation. Other improvements can be made in consistent formatting between figures, larger scalebars, consistent font types however these did not factor into my decision.

Round 3

Reviewer 2 Report

The manuscript has now been improved to meet the standard of Micromachines.

I have two minor comments:

  1. Shall the title of the paper be revised? Because Authors have claimed that they used both EWOD and DEP.
  2. Shall Table 1 be removed? Comparison to only two refs. is somehow weak to show.

Reviewer 3 Report

The manuscript is significantly improved. The remaining issues I have are largely related to editing and improving logical connection between sections. A few suggestions are given but not exhaustive.

38-Please consider a transition sentence between the first paragraph and second. For example, “While printing technology is important a second application area of interest to us is Structural Electronics. Structural Electronics (SE) is a component…”

48-To address applications such as those we have highlighted; we examine the use of a microfluidic technology to manipulate liquids into arbitrary shapes using digitally addressed electrodes. In recent years …

Having added “Theoretical basis of the electro-wetting effect, experimental design and setup” a transition before 2.1 could improve its relevance. Here is an example: “Therefore, a lower limit on the size of a droplet that can be driven is not known and the force applied is simply proportional to the voltage applied. These are important considerations for the application of this approach to high fidelity digital printing at high rates.”

Table 2 is never referenced.

All the scale bars are difficult to read when printed out and are not referenced in the legends.

Example lines that need grammatical editing:

30-34 Very complicated sentence.

34 Therefore, a maskless patterning technology will be of great assistance to early-stage R&D of products, and prevent master mold degradation.

36 In contrast, a mold that can be digitally patterned will not degrade when used to print on a substrate.

41 After a structure has been formed the conductive trace can be patterned on its surface by inkjet...

42-43 At present conductive traces cannot be embedded into the SE component because the structural fabrication and trace patterning occur as serial processes.

43 two separate processes

44 – combined to combine, providing to provide

Finally, it still isn't clear to me that after the first layer when 3D printing that surface tension is all that's needed for the examples demonstrated. However, I appreciate the additional details.

I want to thank the authors for taking the time and effort to revise the paper in detail which I believe has strengthened it significantly. 
